# Relationship between GABA-Ergic System and the Expression of Mephedrone-Induced Reward in Rats—Behavioral, Chromatographic and In Vivo Imaging Study

**DOI:** 10.3390/ijms24129958

**Published:** 2023-06-09

**Authors:** Olga Wronikowska-Denysiuk, Agnieszka Michalak, Anna Pankowska, Łukasz Kurach, Paulina Kozioł, Artur Łazorczyk, Katarzyna Kochalska, Katarzyna Targowska-Duda, Anna Boguszewska-Czubara, Barbara Budzyńska

**Affiliations:** 1Independent Laboratory of Behavioral Studies, Chair of Biomedical Sciences, Medical University of Lublin, Chodzki 4a Street, 20-093 Lublin, Poland; olga.wronikowska-denysiuk@umlub.pl (O.W.-D.); agnieszka.michalak@ulmub.pl (A.M.); lukasz.kurach@umlub.pl (Ł.K.); 2Department of Radiography, Medical University of Lublin, Staszica 16 Street, 20-081 Lublin, Polandartur.lazorczyk@umlub.pl (A.Ł.); katarzyna.kochalska@umlub.pl (K.K.); 3Department of Biopharmacy, Medical University of Lublin, Chodzki 4a Street, 20-093 Lublin, Poland; 4Department of Medical Chemistry, Medical University of Lublin, Chodzki 4a Street, 20-093 Lublin, Poland; anna.boguszewska-czubara@umlub.pl

**Keywords:** addiction, new psychoactive substances, cathinones, conditioned place preference, GABA

## Abstract

Mephedrone is a psychoactive drug that increases dopamine, serotonin and noradrenaline levels in the central nervous system via interaction with transporters or monoamines. The aim of the presented study was to assess the role of the GABA-ergic system in the expression of mephedrone-induced reward. For this purpose, we conducted (a) a behavioral evaluation of the impact of baclofen (a GABA_B_ receptors agonist) and GS39783 (a positive allosteric modulator of GABA_B_ receptors) on the expression of mephedrone-induced conditioned place preference (CPP) in rats, (b) an ex vivo chromatographic determination of the GABA level in the hippocampi of rats subchronically treated with mephedrone and (c) an in vivo evaluation of GABA hippocampal concentration in rats subchronically administered with mephedrone using magnetic resonance spectroscopy (MRS). The results show that GS39783 (but not baclofen) blocked the expression of CPP induced by (20 mg/kg of) mephedrone. The behavioral effect was consistent with chromatographic analysis, which showed that mephedrone (5 and 20 mg/kg) led to a decrease in GABA hippocampal concentration. Altogether, the presented study provides a new insight into the involvement of the GABA-ergic system in the rewarding effects of mephedrone, implying that those effects are at least partially mediated through GABA_B_ receptors, which suggests their potential role as new targets for the pharmacological management of mephedrone use disorder.

## 1. Introduction

According to the newest European Monitoring Centre for Drugs and Drug Addiction (EMCDDA) report, synthetic cathinones are the second largest group of the overused novel psychoactive substances (NPS) [1]. The main representative of this group is mephedrone (4-methylmethcathinone), also recognized as 4-MMC, M-CAT, meow meow or bubbles. Although it reached its peak of popularity about a decade ago, it is still not only one of the most frequently overused drugs of abuse, but it has also been a prototype for the synthesis of new, structurally similar agents with unknown pharmacological and toxicological properties [2,3]. Mephedrone exhibits its effects by increasing the concentration (and therefore, enhancing the activity) of dopamine (DA), noradrenaline (NA) and serotonin (5-HT) in the central nervous system (CNS), mainly due to its interaction with plasma membrane transporters for monoamines [4,5,6,7,8]. Furthermore, it also increases 5-HT and DA concentrations in the nucleus accumbens (NAc) [9,10], frontal cortex and striatum of rats [9]. Recent studies also indicated that mephedrone-induced changes in 5-HT, 5-hydroxy-3-acetic acid (5-HIAA), NA and glutamate levels in different brain regions (the hippocampus, prefrontal cortex (PFC), amygdala, striatum, hypothalamus and thalamus) are strictly dependent on the time of drug treatment in both sexes of rats, suggesting that mephedrone abuse might lead to different neurochemical changes in the process of the transition from occasional to habitual use [11,12]. In our previous study, we also shed light on glutamatergic involvement in mephedrone-induced reward [13]; however, the relationship between GABA-ergic neurotransmission and the rewarding effects of mephedrone remain underexplored.

γ-aminobutyric acid (GABA) is a major inhibitory neurotransmitter in the CNS, mainly acting via two different types of receptors: GABA_A_ and GABA_B_. GABA_A_ receptors belong to ionotropic ligand-gated Cl^−^ channels and are responsible for transmitting fast inhibitory signals that produce an increase in the permeability of the neuronal membrane for Cl^−^ ions [14]. GABA_B_ receptors represent the group of G protein-coupled receptors (GPCRs) and are expressed both pre- and post-synaptically, where they transmit slow and prolonged signals [15]. Ligands acting via both GABA_A_ and GABA_B_ receptors are clinically important in the treatment of several CNS disorders, such as insomnia, epilepsy, muscle spasticity, neuropathic pain, depression, anxiety and drug abuse [15,16]. Notably, available data suggest that GABA_B_ receptors are the ones that are vastly involved in drug abuse disorder [17,18,19] and, importantly, that GABA_A_ receptors may not be crucial in the development of mephedrone-induced reward [20].

The relationship between GABA-ergic neurons and different neurotransmitter pathways in the CNS is a complex issue. Although many neurons utilize GABA as their primary neurotransmitter, a significant amount of GABA-ergic neurons are interneurons and, therefore, can modify the excitability and neurotransmission of other overlapping circuits [14]. The involvement of GABA-ergic transmission in the reward circuit (mainly including the ventral tegmental area (VTA), NAc, PFC, hippocampus or amygdala) has been described thoroughly and in detail [17,18,21]. Briefly, VTA DA neurons are the key players in drug reward that can project to other regions, mediating memory processes, emotional response and self-regulation in reward-related behavior [21,22,23]. Importantly, both GABA_A_ and GABA_B_ receptors are expressed on VTA DA and GABA neurons [24]. Furthermore, VTA DA neurons receive excitatory glutamatergic inputs, which are counterbalanced by inhibitory GABA-ergic inputs from VTA GABA-ergic neurons and from afferents from different subcortical regions [18,25]. This wide range of GABA-ergic inputs to the VTA can activate (1) GABA receptors on VTA DA neurons, which leads to their inhibition, and (2) GABA neurons, which disinhibit VTA DA neurons [18]. Therefore, it can be suggested that these GABA-related mechanisms take part in drug-induced abuse and complement direct and well-established DA-dependent mechanisms, especially so considering the most recent studies on the GABA-ergic system and drug of abuse, which have reported that voluntary alcohol intake in rats can alter the activity of DA neurons in VTA via functional and structural changes in GABA-ergic transmission [26] and that alcohol and cocaine coadministration affects the gene expression involved in the GABA-ergic system in the medial PFC and amygdala [27].

Based on the results obtained in our previous studies on glutamate involvement in mephedrone-induced reward [13] and knowledge of the GABA/glutamate equilibrium, we hypothesized that GABA-ergic neurotransmission can also be involved in the rewarding effects of this drug. Ample evidence has been provided that the GABA-ergic system is involved in the actions of different drugs of abuse and that drugs of abuse can disrupt the GABA/glutamate equilibrium by increasing glutamate (excitatory) transmission and decreasing GABA (inhibitory) transmission (see Section 3). With the aim of verifying this hypothesis, in the presented research, we propose an integrative approach to evaluate the GABA-ergic involvement in the expression of mephedrone-induced rewarding effects.

For this purpose, we performed in vivo and ex vivo determinations of GABA hippocampal levels using magnetic resonance spectroscopy (MRS) and ion-exchange chromatography. To complement these analyses, GABA level was also evaluated in the PFC. The above-mentioned MRS and chromatographic techniques were combined with a behavioral assay in which we aimed to establish whether the administration of ligands acting via GABA_B_ receptors—baclofen (a GABA_B_ receptors agonist) and GS39783 (a positive allosteric modulator (PAM) of GABA_B_ receptors)—are able to block the expression of mephedrone-induced CPP (Figure 1). The above-mentioned blockage of the expression of the previously acquired mephedrone-CPP was assessed by administering the studied GABA-ergic compounds acutely before a post-conditioning test. Altogether, the results presented in this paper provide a novel insight into the relationship between the GABA-ergic system and the expression of the rewarding effects of mephedrone in rats.

## 2. Results

### 2.1. Impact of Baclofen on the Expression of Mephedrone-Induced CPP and on Animals’ Locomotor Activity

Figure 2A shows the impact of baclofen on the expression of mephedrone-induced CPP [two-way ANOVA: baclofen post-treatment: F (2, 38) = 1.888; *p* = 0.1653; mephedrone pre-treatment: F (1, 38) = 7.998; *p* = 0.0074; interaction: baclofen post-treatment × mephedrone pre-treatment: F (2, 38) = 1.656; *p* = 0.2043]. The results show the significance of mephedrone conditioning, and a post hoc Tukey’s test was used to confirm that mephedrone (20 mg/kg) induced CPP (unlike saline-injected rats) (*p* = 0.0296). The administration of baclofen (1 and 3 mg/kg) during the post-conditioning test day did not affect the score values in mephedrone- or saline-conditioned animals.

The effect of baclofen treatment on animals’ mobility in saline- and mephedrone-conditioned animals is shown in Figure 2B. Statistical analysis of the data showed the significant impact of baclofen treatment on rats’ mobility: [two-way ANOVA: baclofen post-treatment: F (2, 38) = 31.06; *p* < 0.0001; mephedrone pre-treatment: F (1, 38) = 0.6523; *p* = 0.4243; interaction: baclofen post-treatment × mephedrone pre-treatment: F (2, 38) = 1.496; *p* = 0.2370]. The post hoc Tukey’s test revealed that post-treatment with baclofen (1 mg/kg) significantly decreased the distance traveled by the animals during the post-conditioning test in both groups of rats, i.e., those conditioned with saline and mephedrone (unlike those animals conditioned with saline and mephedrone but administered with saline during the post-conditioning test (*p* = 0.0119; *p* = 0.0102, respectively)). Baclofen, in a higher administered dose (3 mg/kg), also decreased the animals’ mobility in both groups, i.e., those conditioned with saline and mephedrone (unlike those animals conditioned with saline and mephedrone but administered with saline during the post-conditioning test (*p* < 0.0001; *p* = 0.0012, respectively)).

### 2.2. Impact of GS39783 on the Expression of Mephedrone-Induced CPP and on Animals’s Locomotor Activity

The impact of GS39783 on the expression of mephedrone-induced CPP in rats is presented in Figure 3A. The two-way ANOVA analysis showed the statistically significant effect of the GS39783 post-treatment and the interaction between GS39783 post-treatment and mephedrone pre-treatment: [two-way ANOVA: GS39783 post-treatment: F (3, 56) = 3.531; *p* = 0.0205; mephedrone pre-treatment: F (1, 56) = 0.0004751; *p* = 0.9827; interaction: GS39783 post-treatment × mephedrone pre-treatment: F (3, 56) = 8.382; *p* = 0.0001]. As previously reported, mephedrone (20 mg/kg) produced CPP in animals, unlike those conditioned with saline (post hoc Tukey’s test, *p* = 0.015). Additionally, as revealed using a post hoc Tukey’s test, the administration of GS39783 at a dose of 2.5 mg/kg during the post-conditioning test blocked mephedrone CPP (unlike animals conditioned with mephedrone and injected with saline during the post-conditioning test) (*p* = 0.0043). Treatment with GS39783 at a higher dose (5 mg/kg) also decreased the score values in rats conditioned with mephedrone, unlike animals conditioned with mephedrone and injected with saline during the post-conditioning test (*p* < 0.0001), and unlike animals conditioned with saline and injected with GS39783 (5 mg/kg) during the post-conditioning test (*p* = 0.0254). The results of the treatment with GS39783 at the lowest dose (1 mg/kg) were insignificant. An alternative presentation of the CPP results, shown as the percentage of the time spent in the drug-paired compartment during the post-conditioning test, is presented in the Appendix A.

Figure 3B indicates the impact of the applied protocol and treatment on the animals’ locomotor activity measured during the post-conditioning test. The two-way ANOVA indicated the statistically significant impact of the mephedrone-pretreatment on the animals’ mobility: [two-way ANOVA: GS39783 post-treatment: F (3, 56) = 0.4873; *p* = 0.6925; mephedrone pre-treatment: F (1, 56) = 10.01; *p* = 0.0025; interaction: GS39783 post-treatment × mephedrone pre-treatment: F (3, 56) = 0.6743; *p* = 0.5714]; nevertheless, the post hoc Tukey’s test with multiple comparisons did not show statistically significant differences between any of the tested groups.

### 2.3. Chromatographic Determination of GABA Hippocampal/PFC Concentrations

The effects of mephedrone (5, 10 and 20 mg/kg) administration on GABA concentration in the hippocampus and PFC, measured using ion-exchange chromatography, are presented in Figure 4A,B, respectively.

Statistical analysis the showed significant effect of 6 days of mephedrone treatment on the GABA hippocampal level when measured 24 h after the last mephedrone injection: [one-way ANOVA: F (3, 28) = 6.052; *p* = 0.0026]. Furthermore, the post hoc Tukey’s test showed that the subchronic administration of mephedrone (5 and 20 mg/kg) significantly decreased the GABA level in rats’ hippocampi (unlike saline-treated animals (*p* = 0.024; *p* = 0.0424, respectively)). Statistical analysis also showed the significant effect of 6 days of mephedrone treatment on the GABA PFC level when measured 24 h after the last mephedrone injection: [one-way ANOVA: F (3, 16) = 3.275; *p* = 0.0485]. Furthermore, the post hoc Tukey’s test showed that the subchronic administration of mephedrone (5 mg/kg) significantly decreased the GABA level in rats’ PFC (unlike saline-treated animals (*p* = 0.0315)).

The analytical curve constructed for GABA was linear in the tested range. The value of the determination coefficient was calculated (r2 = 0.999, y = 0.068x + 3.7328). The LOD and LOQ determined were 38 nM and 114 nM, respectively. The representative chromatograms registered for the hippocampi of the rats treated with saline and mephedrone (5, 10 and 20 mg/kg) are presented in Figure 5. An exemplary chromatogram of the standard solution at a concentration of 250 nM is presented in the Appendix A (Appendix A).

### 2.4. MRI Analysis of GABA Hippocampal Level

The results presented in Figure 6C,D show the GABA hippocampal level measured using MRS 24 h and 2 weeks after the last mephedrone injection. Statistical analysis did not show any significant differences in GABA concentration in either of the tested time points of the measurements taken after the last mephedrone injection: [one-way ANOVA: F (3, 28) = 1.623; *p* = 0.2063 (after 24 h); one-way ANOVA: F (3, 28) = 1.561; *p* = 0.2208 (after 2 weeks)].

## 3. Discussion

The presented research was undertaken to evaluate the relationship between the GABA-ergic system and the expression of the rewarding effects of mephedrone in rats. For this purpose, an integrative approach was proposed, which consisted of a behavioral study assessing the impact of the GABA_B_ receptors agonist and the PAM of the GABA_B_ receptors on the expression of mephedrone-induced CPP, and the chromatographic and MRS determination of GABA concentration measured in the hippocampi of mephedrone-treated rats.

Literature data on the relationship between mephedrone effects and the GABA-ergic system are scarce and mostly related to GABA_A_ receptors. It has been reported that mephedrone can induce CPP in both wild type mice (δ-WT mice) and in knock-out mice lacking the δ subunit-containing GABA_A_ receptor (δ-KO mice), whereas morphine has been found to be able to induce CPP only in δ-WT mice [20]. This may suggest that δ-GABA_A_ receptors are crucial for opioid-induced reward, presumably due to the activation of presynaptic μ-opioid receptors and inhibition of GABA release in the VTA with the subsequent disinhibition of the VTA DA neurons [20,28]. Since mephedrone was able to induce CPP in both genotypes (δ-WT and δ-KO mice), it can be concluded that GABA_A_ receptors may only partially take part in the mephedrone-mediated response, but they are not crucial in the development of mephedrone-induced reward.

The available data suggest that GABA_B_ receptors are the ones that are most strongly considered to be molecular targets for substance use disorder treatment [17,18,19]. Furthermore, most GABA_A_ receptor agonists display anxiolytic properties and can exhibit addictive potential [29], thus making their use in drug abuse treatment non-preferential. Therefore, taking into consideration the above-mentioned reasons, in our research, we focused on the comprehensive evaluation of GABA_B_ receptors’ involvement in the rewarding effects of mephedrone. For this purpose, we utilized two agents acting via GABA_B_ receptors that have been previously shown to reduce the reward induced by different drugs of abuse: baclofen (GABA_B_ receptor agonist) and GS39783 (PAM of GABA_B_ receptors).

Baclofen has been used as a tool drug in addiction studies for over two decades (see [30] for review) and has been shown to reduce the reinforcing and rewarding effects of cocaine [31,32], nicotine [33,34], heroin [35,36], amphetamine [37] and alcohol [38,39] in several behavioral assays in rodents. Furthermore, it has been reported that pretreatment with baclofen reduces nicotine-, morphine- and cocaine-evoked DA release in the shell of the NAc [40]. Interestingly, baclofen has also been used in the treatment of alcohol dependence or alcohol use disorders in humans [41]; however, the data regarding this are insufficient to elaborate on this subject.

In our research, baclofen showed a dose-dependent effect in mephedrone-conditioned animals; however, statistically, it failed to block the expression of mephedrone-induced CPP. Nevertheless, the underlying cause of the lack of this blockage is most likely attributed to the baclofen-induced decrease in animals’ locomotor activity. In order to make sure that the lack of baclofen effect in CPP is caused by the myorelaxation (which leads to the received data distribution), we performed a post hoc power analysis that showed a non-satisfactory level of power for baclofen post-treatment and the interaction of baclofen post-treatment and mephedrone pre-treatment (Appendix A).With an increase in group size, the baclofen effect would probably reach statistical significance; however, an increase in sample size was not possible in the present study due to ethics reasons regarding the use of animals in research. Since baclofen is clinically widely used as a myorelaxant [42], we were aware that a reduction in rats’ mobility might occur; nevertheless, based on the numerous studies published on baclofen use in drug abuse research and in people with drug addictions, we decided to administer this compound. Although, in this study, we report a mephedrone-induced reduction in GABA concentration in the hippocampus, based on baclofen-related findings, we could not fully confirm the involvement of GABA_B_ receptors in the rewarding effects of mephedrone. Furthermore, its therapeutic potential in drug abuse disorders is somewhat limited by its myorelaxant and sedative properties [42,43]. Therefore, to properly assess GABA_B_ receptors’ involvement in the rewarding effects of mephedrone, we also tested the PAM of these receptors: GS39783.

GS39783 is a promising agent that has been repeatedly used in research on drug abuse using animal models. GS39783 has been shown to attenuate the alcohol-induced increase in locomotion and potentiate the induction of locomotor sensitization [44], as well as being shown to inhibit alcohol self-administration [45,46,47] and suppress the acquisition of alcohol drinking behavior [48]. The administration of GS39783 has also been shown to attenuate cocaine-induced hyperlocomotion [49] and reduce its reinforcing effects in self-administration [50] and intracranial self-stimulation (ICSS) [32]. Furthermore, it has been shown to successfully block the expression of amphetamine- [50] and methamphetamine-induced CPP [51]. Lastly, GS39783 has been shown to be able to decrease nicotine self-administration [52] and block the acquisition (but not the expression) of nicotine reward in the CPP paradigm [53].

Ample evidence has proven the ability of GS39783 to block the rewarding and reinforcing effects of different drugs of abuse. Its utility was confirmed by our study, in which GS39783 (2.5 and 5 mg/kg) blocked the expression of mephedrone-induced CPP. This shows that GABA_B_ receptors are involved in the rewarding effects of mephedrone and that the PAM of these receptors can be successfully used to reduce the mephedrone-induced reward. Interestingly, for a higher dose of GS39783 (5 mg/kg), an aversive behavior was found. We hypothesize that this effect might have been seen due to the inhibitory effect of GABA VTA neurons on DA VTA neurons [54], which may result in a decrease in DA level and place aversion [55]. At the same time, a medium dose of GS39783 (2.5 mg/kg) showed a classic inhibition of mephedrone-induced CPP without any aversive effect, and for this reason, this dose should be seen as the most prominent in the presented study. It is also worth mentioning that, in terms of GABA_B_ receptors activity, PAMs can serve as a safer alternative to orthosteric binding ligands as they possess the ability to potentiate the action of endogenous agonists and modulate the receptor signaling without replacing the endogenous ligands from their binding site [15]. The beneficial effects of PAMs of GABA_B_ receptors compared to classic GABA_B_ receptors agonists are also reflected in the lack of their myorelaxant activity.

It is worth mentioning that the studied GABA-ergic compounds were administered acutely before the post-conditioning test, which was performed 24 h after the last mephedrone injection. The literature data from animal models suggest that (a) the half-time (t ½) of mephedrone after intravenous (i.v.) injection is short and equals 0.37 h, and (b) mephedrone is undetectable in plasma 9 h after oral administration [56]. Although, in our study, mephedrone was administered via i.p. injections, we hypothesize that the concentrations of plasma are similar to the above-mentioned i.v. route of administration. Therefore, we conclude that neither baclofen nor GS39783 (administered 24 h after the last mephedrone injection) interacted directly with mephedrone; however, GS39783 was able to block the expression of the previously acquired mephedrone-CPP, presumably due to mephedrone-induced changes in the central GABA concentration level.

In the presented study, the subchronic mephedrone treatment did not affect the animals’ locomotor activity when measured 24 h after the last mephedrone injection. In our previous studies, we also did not observe any changes in animals’ mobility after this measurement time after administration [13,57]. Although other studies have shown an increase in animals’ locomotion after acute [6,56,58,59,60] and repeated [61,62,63] mephedrone treatment, this increase has been observed when tested immediately after drug administration and lasted 40–120 min [56,58,64]. The lack of these changes in the animals’ mobility being observed in the presented research was not only predictable but also desirable, and therefore, we may conclude that mephedrone did not affect the measured baclofen- and GS39783-related effects.

Our study is the first study that evaluates GABA concentration in the CNS, specifically in the hippocampus and PFC, following mephedrone treatment. Importantly, it has been reported that in humans with addictions to cocaine and/or alcohol, as well as in alcohol-preferring rats, a reduction in GABA_B1_R mRNA expression in the hippocampus was observed [65], further supporting the hypothesis of the utility of the GABA_B_ receptors’ ligands in drug abuse treatment. Furthermore, the hippocampus receives projections of DA neurons from the VTA and therefore plays a significant role in drug reward circuits by mediating the memory processes and emotional response [21,22]. PFC also plays an important role in the drug reward system as it is able to enhance drug reward by projecting to other mesocorticolimbic regions and receiving projections from midbrain DA neurons [66]. It is also responsible for processing emotional information, integrating drug-related decision making and regulating inhibitory inputs [67]. Interestingly, GABA-ergic inhibitory interneurons represent approximately 10–15% of the total neuron population in the hippocampus [68]; nevertheless, they are crucial in regulating excitatory glutamate inputs [69]. In the light of our previous results showing that mephedrone administration increases the glutamate hippocampal level [13], the determination of GABA hippocampal concentration was crucial to complement these findings.

For this purpose, two corresponding methods were used (ex vivo and in vivo). In vivo spectroscopy imaging allowed the assessment of time-dependent changes in the same cohort of animals in a non-invasive way; thus, it is an advantageous method for ex vivo chromatographic determination. Although its validity in assessing the glutamate hippocampal level was proven in our previous study [13], here, it did not reveal any significant changes in GABA concentration in the hippocampus in either of the measurement time points (24 h and 2 weeks after the last mephedrone administration). The lack of observed changes may be due to signal overlap from more abundant metabolites such as creatine at 3 ppm; glutamine and glutamate, often denoted as Glx at 2.3 ppm; and N-acetyl aspartate at 2 ppm [70]. GABA exists in rodent brains at a relatively low concentration (less than 2 mM in the hippocampus) [71] and presents broad peaks with a low amplitude in the MRS spectrum. The results obtained in the concentration analysis are likely to have been contaminated by signals from different neighboring compounds [72]; however, the effect of signals overlap is less pronounced in higher fields due to, inter alia, greater spectral resolution and frequency range [72,73,74]. Therefore, an alternative analytical method is proposed.

The ex vivo determination performed using ion-exchange chromatography showed that 6 days of daily mephedrone administration (5 and 20 mg/kg) resulted in a decrease in GABA hippocampal concentration, when measured 24 h after the last mephedrone injection. This reduction, along with the consistent behavioral effects of the GS39783-induced blockage of mephedrone-induced CPP, prove that the GABA-ergic system (specifically, GABA_B_ receptors) is involved in the expression of mephedrone-rewarding effects. For the sake of clarity and transparency, the observed results should be discussed with regard to our previous study, in which mephedrone induced a rewarding effect in the CPP at the doses of 10 and 20 but not 5 and 30 mg/kg [57]. Surprisingly, in the presented research, we did not observe a decrease in the GABA hippocampal level after the administration of mephedrone at the dose of 10 mg/kg. We have observed a similar pattern previously [13]. Since GABA neurons in the brain reward system often serve as interneurons [14], the observed changes could have resulted from alterations in other neurotransmitters’ levels; however, further research and clarification are required to elucidate the exact mechanisms. Finally, it should be mentioned that, in our studies, the administration of mephedrone at a low dose of 5 mg/kg (that did not produce a rewarding effect in the CPP in our previous research) led to a decrease in GABA hippocampal/PFC in the presented paper and an increase in glutamate [13] hippocampal level. This effect was not observed in the PFC for higher doses of mephedrone (10 and 20 mg/kg), which may suggest the lack of involvement of this region in mephedrone-induced reward or interactions with other neurotransmitters. Nonetheless, taking into account that a decrease in GABA hippocampal/PFC level and an increase in glutamate hippocampal [13] level were observed after mephedrone treatment with a dose that did not trigger reward, this may suggest that both of these systems are involved in mephedrone-induced reward; however, further studies should be conducted in order to better understand the exact mechanism of mephedrone-induced reward and its associated consequences. Nonetheless, both of our studies—the present study and the previous one—suggest that in order to observe central changes in the neurotransmitters’ level, the use of two complementary methods is a valid and sometimes inevitable procedure.

Although the research presented in this paper shows interesting outcomes on GABA-ergic involvement in the expression of mephedrone-induced reward, we identified a few limitations that should be considered and addressed in future studies. Firstly, we are aware that studying sex-dependent changes would add value to the presented research. Secondly, we are aware that performing CPP in batches has its limitations, especially when not all of the tested doses are assigned to all of the performed batches. However, we decided to use this design due to the fact that it was only after performing the first batch that the decision was made on whether to administer higher or lower doses of the tested compounds. To make the results reliable, control animals were added to each of the tested batches, and all efforts possible were made to make the experimental conditions equal for each batch. Furthermore, the evaluation of the central concentrations of GABA was performed in the hippocampus and PFC. Presumably, the assessment of GABA level in other areas of the drug reward system (such as the VTA or NAc) would add informational value to the presented research. Furthermore, we failed to observe changes in GABA concentration using MRS. The protocol applied in our study was chosen in order to achieve a high signal-to-noise ratio (SNR) and obtain reliable results from all metabolites existing in the spectrum. On the one hand, it could be worth considering the application of GABA-edited solutions such as MEGA-PRESS [75] or the J-resolved [76] method. On the other hand, in the presented research, we took into account the fact that the adjustments mentioned above might compromise the results. Finally, we failed to observe changes in GABA hippocampal level following the administration of mephedrone at a dose of 10 mg/kg in both of the applied analytical methods. Without additional analysis, this enigmatic effect remains in the sphere of speculation. Therefore, further studies are of great importance in order to fully explain the relationship between mephedrone administration and GABA-ergic transmission in the hippocampus.

Altogether, the presented research provides novel information on the involvement of the GABA-ergic system in the expression of mephedrone-triggered reward. The mephedrone-induced decrease in the GABA hippocampal concentration was consistent with the behavioral effect, where the PAM of the GABA_B_ receptors, GS39783, blocked the expression of mephedrone-induced CPP. These findings suggest that the rewarding effects of mephedrone are at least partially mediated through GABA_B_ receptors. This, along with the abundance of existing evidence on PAMs of GABA_B_ receptors’ utility in drug abuse-related research, strongly suggest that they may offer a novel therapeutic approach for the treatment of substance use disorder. Furthermore, the findings obtained in the presented study complement our previous results of glutamate involvement in the rewarding effects of mephedrone [13], and together, the results of our previous study and the research presented prove that both the GABA-ergic and glutamatergic systems are implicated in the expression of mephedrone-induced reward. This not only sheds light on new therapeutic options for mephedrone-induced abuse and addiction but also provides informational value regarding the possible involvement of these systems in other cathinone derivatives’ mechanisms of action. Although more studies are needed to confirm this hypothesis, the amount of new NPS from the cathinone derivatives group is enormous, and therefore, it would not always be possible and feasible to test each of the new synthesized compounds. Thus, in many cases, knowledge on mephedrone, a prototype drug from this group, would probably be extrapolated to other cathinone derivatives.

## 4. Materials and Methods

### 4.1. Animals

All experiments were performed on drug-naive male Wistar rats (approximately 8 weeks old, weighing 200–250 g at the beginning of the experiments) and were conducted between 8.30 a.m. and 4.30 p.m. Rats were obtained from the Centre of Experimental Medicine of the Medical University of Lublin and were housed in pairs with sex-matched and weight-similar conspecifics. The experimental groups consisted of 7–8 rats (detailed numbers are presented in figure legends). The animals were kept and maintained under standard laboratory conditions that included (a) a 12 h light/dark cycle with lights at 8.00 a.m., (b) a 21 ± 1 °C room temperature and (c) a relative humidity of 50 ± 5%). All the time (with the only exception of the testing time), the animals had ad libitum access to laboratory chow (Agropol, Motycz, Poland) and water.

### 4.2. Ethics Statement

All experiments were (a) performed in compliance with ARRIVE guidelines, (b) carried out in accordance the National Institute of Health Guidelines for the Care and Use of Laboratory Animals and to the European Community Council Directive for the Care and Use of Laboratory Animals of 22 September 2010 (2010/63/EU) and (c) approved by the Local Ethics Committee in Lublin, Poland (Permission No: 53/2018).

### 4.3. Drugs

In the described experiments, the following compounds were utilized: mephedrone hydrochloride (Tocris Bioscience, Bristol, UK, Cat. No. 4443), baclofen (Sigma Aldrich, Burghausen, Germany, Cat. No. B5399) and GS39783 (Sigma Aldrich, Burghausen, Germany, Cat. No. G5919). For the CPP protocol, mephedrone was injected once daily (during the afternoon session) for 6 days at a dose of 20 mg/kg. This dose was chosen based on our previous study, where mephedrone at this dose produced the most pronounced response in the CPP [57]. Baclofen (at doses of 1 and 3 mg/kg) and GS39783 (at doses of 1, 2.5 and 5 mg/kg) were injected acutely 30 min prior to the post-conditioning test. The chosen range of doses and time of administration were based on our preliminary studies and other available data [33,50,77]. In order to assess the possible mephedrone-induced dose-dependent changes in GABA hippocampal concentrations, for the in vivo MRI study and ex vivo chromatographic analysis, mephedrone was administered to different cohorts of animals at doses of 5, 10 and 20 mg/kg for 6 consecutive days. Ex vivo chromatographic analysis was carried out in the animals’ hippocampi dissected 24 h after the last mephedrone injection. In vivo MRI was performed 24 h and 2 weeks after the last mephedrone treatment. Different cohorts were used for the CPP, chromatographic and MRS studies. All drug solutions used were prepared by dissolving substances in 0.9% sterile saline. An exception was made for GS39783, which was suspended in 0.9% sterile saline with a drop of Tween 80 (creating a solution with a maximum of 0.25% of Tween 80 concentration). All drugs used were injected intraperitoneally (i.p.) at a volume of 2 mL/kg. The control groups were treated with 0.9% sterile saline (i.p.; 2 mL/kg). For the MRI experiments, the animals received an anesthetic mixture that consisted of 3.5% isoflurane and 100% oxygen. The flow level of an anesthetic mixture was set at 0.7 L/min. The animals were kept under anesthesia during the whole scan with a dose of isoflurane ±2% (the lowest possible).

### 4.4. Experimental Procedure and Treatment

#### 4.4.1. CPP Paradigm

CPP apparatus, software and groups assignment

The CPP test was performed in Ugo Basile two-compartment CPP boxes separated with guillotine doors. Each compartment was a square and 30 × 30 × 30 cm in size. The compartments differed by wall colors/patterns and floor grid (black walls with square 0.1 × 0.1 cm holes in the floor vs. black and white striped walls with round 0.2 cm holes in the floor). This set of compartments was previously validated in our laboratory [13,56] as a set for unbiased protocol (the initial preference for both compartments is similar); thus, the animals were randomized and half on them (in each experimental group) was conditioned in the “black” compartment and the other half was conditioned in the “striped” compartment. In order to be qualified for an unbiased protocol, animals had to show a preference for each context in the maximum range of 40–60%. The measurements of the initial preference are presented in the Appendix A (Appendix A and Appendix A).

The experiments were performed in six CPP boxes, and therefore, six animals were tested simultaneously. Due to the significant number of the animals used, the experiments presented in this paper were performed in batches: (a) the first batch included saline + saline (n = 4), mephedrone + saline (n = 4), saline + baclofen 3 mg/kg (n = 7) and mephedrone + baclofen 3 mg/kg (n = 7); (b) the second batch included saline + saline (n = 4), mephedrone + saline (n = 4), saline + baclofen 1 mg/kg (n = 7) and mephedrone + baclofen 1 mg/kg (n = 7); (c) the third batch included saline + saline (n = 4), mephedrone + saline (n = 4), saline + GS39783 5 mg/kg (n = 8) and mephedrone + GS39783 5 mg/kg (n = 8); (d) the fourth batch included saline + saline (n = 4), mephedrone + saline (n = 4), saline + GS39783 2.5 and 1 mg/kg (n = 8, n = 8, respectively) and mephedrone + GS39783 2.5 and 1 mg/kg (n = 8, n = 8, respectively).

During the experiments, VideoMot 2 TSE Systems software was utilized to identify the location of the animals in each compartment using the contrast difference (animal vs. floor). This software was used for both (1) the measurement of the time spent by the animals in each compartment and (2) the measurement of the distance traveled by the animals.
CPP test

The CPP test was previously validated in our laboratory and was carried out as described in our previous papers [13,57]. Firstly, before the experiments began, the animals were handled and habituated to the apparatus for 15 min one day prior to the pre-conditioning test. The applied CPP protocol included three stages: (a) pre-conditioning (day 0), (b) conditioning (days 1–6) and (c) post-conditioning (day 7). Throughout the pre-conditioning test, the animals could freely move between the compartments (door opened) while their initial preference was measured for 15 min. During the 6 days of conditioning, the guillotine door was closed. In the first (morning) session, the animals were administered saline and confined in one of the compartments for 30 min. During the second (afternoon) session, the animals were treated with saline (control groups) or mephedrone (20 mg/kg) and confined in the second compartment for 30 min. The time between the morning and afternoon sessions was 4 h. In order to evaluate the impact of the studied GABA-ergic compounds on the expression of mephedrone-induced reward, on the last day of the CPP procedure, the animals underwent a post-conditioning test. Saline (control groups), baclofen (0, 1 or 3 mg/kg) or GS39783 (0, 1, 2.5 or 5 mg/kg) were acutely injected 30 min before the test. The guillotine door was opened and, similarly to the pre-conditioning test, the animals could freely move between two compartments. The post-conditioning preference of the animals was measured for 15 min. The graphical representation of the CPP procedure is shown in Figure 1.

#### 4.4.2. Locomotor Activity

In order to evaluate whether the administered drugs had an impact on the animals’ mobility during the test, the distance moved by the animals was measured for 15 min during the post-conditioning test (24 h after the last mephedrone injection and 30 min after the baclofen or GS39783 injection). As for the CPP, the VideoMot 2 TSE Systems program was used here.

#### 4.4.3. MRS

The MR measurements were performed in accordance with our previous work [13]. Briefly, two separate MR scans were performed in 32 rats (treated with saline or mephedrone at a dose of 5, 10 or 20 mg/kg for 6 days; n = 8). The first scan was performed 24 h after the last mephedrone injection and the second one was performed 2 weeks after the last mephedrone treatment. Food deprivation was implemented for 6 h before each MRI experiment. The breathing rate (maintained at about 40–50 bpm) and body temperature (about 37 °C) were monitored during the whole scan time (about 2.5 h per animal).

MRS was performed on an MR 7T horizontal bore magnet (70/16 Pharma Scan, ParaVision 6.0.1, Bruker Biospin, GmbH, Rheinstetten, Germany) using a 72 mm inner diameter volume coil for transmitting and a 20 mm surface loop coil for receiving. First, for planning purposes, T2-weighted rapid acquisition with refocused echoes (RARE) images were acquired in 3 anatomical planes covering the whole brain area. Subsequently, the proton MRS spectra were measured using the PRESS (Point Resolved Spectroscopy Sequences) in the volume of interest (VOI), with a size of VOI = 0.2 × 0.2 × 0.5 cm (20 μL) placed in the right hippocampus (TR = 2500 ms, TE = 16 ms, averages = 1024, acquisition points = 2048). For detailed sequence parameters and measurements methodology, see [13].

The analysis of MRS spectra was performed using LCModelTM (Linear Combination of Model Spectra) software (version 6.3-1). The analyzing window (0.2–4 ppm) and standard configuration were applied. In order to obtain a reference for the absolute quantification of GABA, a signal of unsuppressed water (from the same VOI as the MRS spectra) was utilized. The GABA signal was extracted for each animal from the whole hippocampal spectrum to calculate the mean spectra in examined groups. The mean spectra of GABA derived from the hippocampus are presented in Figure 6A,B.

#### 4.4.4. Chromatographic Analysis

Following the 6 days of mephedrone (5, 10, 20 mg/kg) treatment (24 h after the last mephedrone injection), the decapitation of the rats was performed by a trained researcher. The brain was removed from the skull and cleared with sterile saline. Subsequently, rats’ hippocampi and PFC were dissected and frozen at a temperature of −80 °C for further use in the chromatographic analysis. All procedures were performed according to the manufacturer’s instructions. Standards and reagents were provided by Ingos Corp. (Prague, Czech Republic). Reagents for buffers preparation (citric acid monohydrate, lithium citrate tribasic tetrahydrate and lithium chloride anhydrous) were purchased from Sigma-Aldrich (Burghausen, Germany). Before the chromatographic procedure, the hippocampi were homogenized and then deproteinized in 6% sulphosalicylic acid in a lithium citrate buffer (pH 2.6) in a ratio of 1:5 and subsequently centrifuged (20 min at 12,000 rpm at 4 °C). The prepared supernatants were utilized for GABA concentration evaluation. For this purpose, an Automatic Amino Acid Analyzer AAA 500 (Ingos Corp., Prague, Czech Republic) based on ion-exchange chromatography with a post-column derivatization of ninhydrin and a UV/VIS detector operated at 440/570 nm was used. For the separation of amino acids, 100 µL of sample was injected onto the OSTION LG FA column (Ingos Corp., Prague, Czech Republic), which was thermostated at 46–80 °C. Mobile phase 5 lithium citrate buffers (with the addition of sodium azide and thiodiglycol) were used. The rate flow of the mobile phase was 0.4 mL/min. The following conditions were applied: 0–45 min for buffer 1 (pH 2.95), 45–69.4 min for buffer 2 (pH 3.1), 69.4–92.9 min for buffer 3 (pH 3.5), 92.9–141.9 min for buffer 4 (pH 4.65), 141.9–150.5 min for 0.5 M of LiOH and 150.5–177.1 min for buffer 5 (pH 2.6). The column temperature program was as follows: 0–43 min at 46 °C, 43–75.8 min at 65 °C, 75.8–161.3 min at 80 °C and 161.3–177.1 min at 46 °C.

The derivative reagent, ninhydrin monohydrate solution supplemented with hydrindantin dihydrate and acetate buffer (pH 5.5), was prepared in 2-methoxyethanol and kept in a nitrogen atmosphere in the dark at +4 °C. The flow rate was 0.2 mL/min. The reactor temperature was set at 120 °C. The autosampler was cooled to +4 °C. The time of one analysis was 177.1 min. Clarity 8.1 software (DataApex, Prague, Czech Republic) was used to calculate the GABA concentration based on 570 nm detection. Linearity was determined by constructing an analytical curve with 7 reference substance concentrations in the range of 40–2500 nM. The peak areas were plotted against the respective concentrations of GABA. The results were subjected to regression analysis using the least squares method to calculate the calibration equation and determination coefficient. The LOD and LOQ for the method were estimated as SD/b ratios of 3.3 and 10, where SD and b represent the standard deviation of the intercept and slope of the calibration line in the LoD region, respectively. The final results of the GABA concentration are presented in mM and the density of the brain tissue in the calculations equaled 1.

#### 4.4.5. Statistical Analysis

For the statistical analyses, GraphPad Prism was used (version 8.0.1). Firstly, the normal distribution of the data was checked using the D’Agostino and Pearson tests. Secondly, depending on the experiment, one- or two-way analysis of variance (ANOVA) with multiple comparisons with a post hoc Tuckey’s test was performed. In the analysis of the CPP results and locomotor activity results, a two-way ANOVA was performed and the mephedrone-/saline-conditioning (pre-treatment) was set as one of the factors, whereas the dose of baclofen or GS39783 (post-treatment) was set as the second factor. For the analysis of the MRS/chromatographic results of the GABA concentration, a one-way ANOVA with multiple comparisons was used. For the calculations of statistical power of the analysis used, IBM SPSS Statistics software (version 29.0.1.0) was used. The data are expressed as (a) the means of scores ± the standard error of mean (SEM) for the CPP results (difference in time (s) spent in the drug-paired compartment during the post- and pre-conditioning tests), (b) the means ± the SEM of the distance moved (m) by animals during the post-conditioning measurement for the locomotion results, (c) the means ± the SEM of the GABA concentrations (mM) for the MRS and (d) the means ± the SEM of the GABA concentrations (mM) for the chromatographic analysis. In all of the statistical calculations, the statistically significant limit of confidence was set at *p* < 0.05.

## Figures and Tables

**Figure 1 ijms-24-09958-f001:**
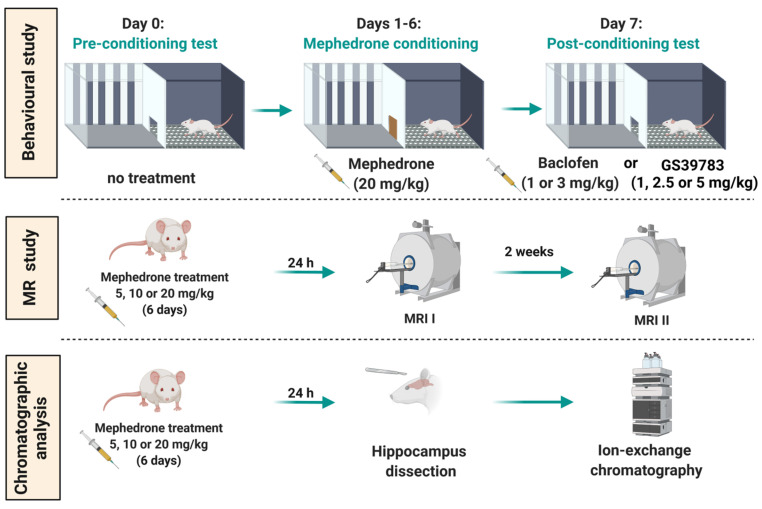
Graphical scheme of the experimental protocol applied in the presented research. This figure was created using https://biorender.com/ (accessed 15 January 2023).

**Figure 2 ijms-24-09958-f002:**
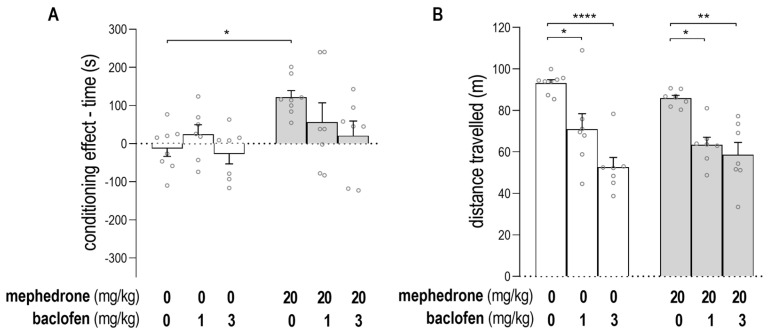
The impact of baclofen on the expression of mephedrone-induced conditioned place preference (CPP) (**A**) and animals’ locomotion (**B**). Data represent means ± SEM and are expressed as (**A**) the difference (in s) in time spent in the drug-paired compartment (post-conditioning–pre-conditioning) and (**B**) the distance moved by the animals (in m) during the post-conditioning test. n = 7 rats per group in the baclofen-treated group and 8 rats per group in the saline-treated group during post-conditioning test. * *p* < 0.05, ** *p* < 0.01; **** *p* < 0.000 (Tukey’s test).

**Figure 3 ijms-24-09958-f003:**
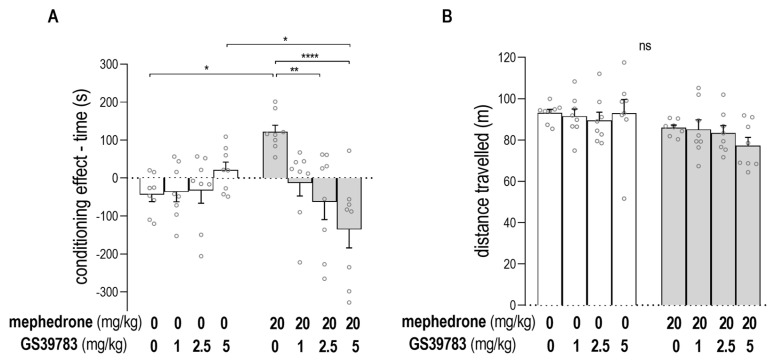
The impact of GS39783 on the expression of mephedrone-induced CPP (**A**) and animals’ locomotion (**B**). Data represent means ± SEM and are expressed as (**A**) the difference (in s) in time spent in the drug-paired compartment (post-conditioning–pre-conditioning) and (**B**) the distance moved by the animals (in m) during the post-conditioning test. n = 8 rats per group. * *p* < 0.05, ** *p* < 0.01; **** *p* < 0.0001; ns—no statistically significant difference (Tukey’s test).

**Figure 4 ijms-24-09958-f004:**
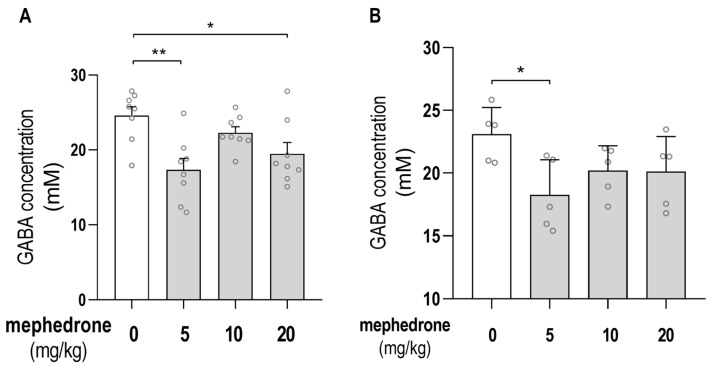
GABA hippocampal (**A**) and PFC (**B**) concentration levels measured using ex vivo ion-exchange chromatography 24 h after the last mephedrone injection. Analyses were performed on the hippocampi of rats administered with mephedrone (5, 10 and 20 mg/kg) for 6 days. n = 8 samples per group for the hippocampus and n = 6 samples per group for the PFC. The data are expressed as means ± SEM of the GABA concentrations (mM/kg tissue). * *p* < 0.05, ** *p* < 0.01 (Tukey’s test).

**Figure 5 ijms-24-09958-f005:**
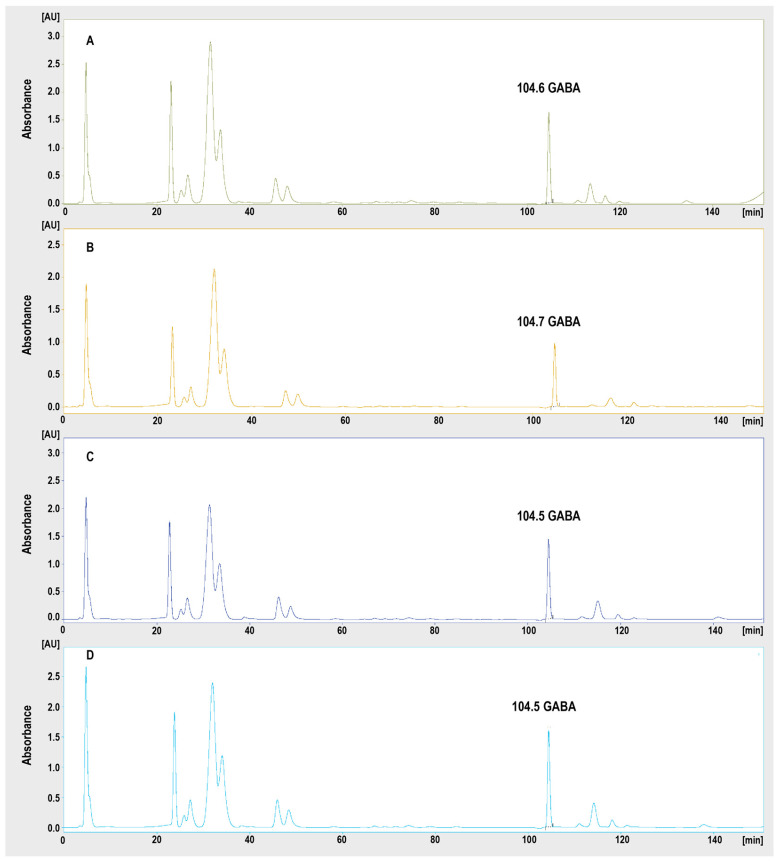
Representative chromatograms registered from the hippocampi of rats treated for 6 days with saline (**A**) or 5 mg/kg (**B**), 10 mg/kg (**C**) or 20 mg/kg (**D**) of mephedrone. The detection wavelength for GABA was 570 nm.

**Figure 6 ijms-24-09958-f006:**
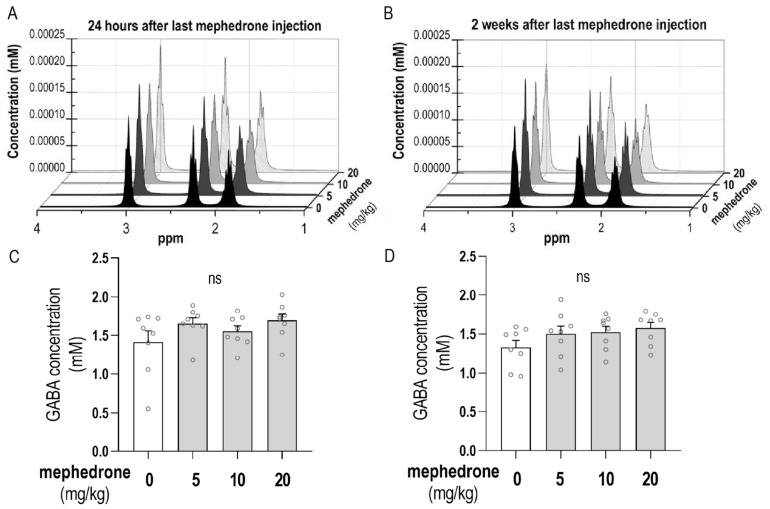
Mean spectra of GABA derived from the hippocampus calculated from the examined groups (**A**) 24 h and (**B**) 2 weeks after the last mephedrone injection. The GABA hippocampal concentration levels measured in vivo using MRS (**C**) 24 h after the last mephedrone injection and (**D**) 2 weeks after the last mephedrone injection. Analyses were performed on rats administered with mephedrone (5, 10 and 20 mg/kg) for 6 days. n = 8 samples per group. The data (**C**,**D**) are expressed as means ± SEM of the GABA concentrations (mM). ns—no statistically significant difference.

## Data Availability

All data generated in the performed experiments are presented in the manuscript. Raw data are available on request.

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
