# Peer review of "Relationship between GABA-Ergic System and the Expression of Mephedrone-Induced Reward in Rats—Behavioral, Chromatographic and In Vivo Imaging Study"

_ijms, 2023, doi:10.3390/ijms24129958_

Round 1

Reviewer 1 Report

The manuscript by Wronikowska-Denysiuk et al reports on the effects of the GABA-B agonist baclofen and the GABA-B PAM on the expression of mephedrone CPP.  They further investigated the effects of mephedrone on GABA levels in the PFC and hippocampus using two different methods.  They found that baclofen had no effect of expression of mephedrone CPP, although it did reduce activity in the expression test.  The PAM GS39783 did reduce expression of mephedrone CPP.  They showed that mephedrone decreased GABA levels in hippocampus at the dose used for CPP using a chromatographic method, but had no effect in PFC.  Using MRS they found no effect of mephedrone on GABA in the hippocampus.  I have a number of concerns with the manuscript.

First, the authors are really studying the effect of the GABA drugs on the memory for a mephedrone CPP and not on mephedrone CPP.  All the central effects of mephedrone they cite in the introduction, such as its effects in the VTA and NAC are not likely to be present during the CPP test when the baclofen drugs are administered.  The hippocampus could be important during the test, but it is unclear to me how that memory effect is specific to mephedrone.  In addition, the one study the authors cite related to GS39783 states that this drug blocked the acquisition, but not expression of nicotine CPP.  It would have made more sense to me to test the GABA drugs on acquisition.

Another concern is the reversal from a preference to what looks like a significant aversion after treatment with GS39783 on the CPP test.  I don’t know what that means for in terms of interpretation of the drug effect or even how that could occur.  Any thoughts?

Some more minor concerns:

The scales for the preference test are different for the two drugs.  These scales should be the same.  The use of different scales makes the baclofen effect look small.

Why weren’t the lower doses of mephedrone tested in the CPP tests?  If you want to make comparison to the neurochemical work, using the same doses would have been helpful.  The largest effects on chromatography were at 5 mg/kg, but that dose wasn’t tested on behavior.  Maybe there was an increase in locomotion at that dose.

What about anesthetic effects on the imaging data?  Is there data to support that isoflurane had no effect?

The specificity of the effects to mephedrone needs to be addressed.  Would the GABA drugs have the same effect on other drug or non-drug CPP?

What would it mean for GABA-B receptors specifically if GABA levels are increased in the hippocampus?

Only minor errors.

Author Response

Reviewer #1

The manuscript by Wronikowska-Denysiuk et al reports on the effects of the GABA-B agonist baclofen and the GABA-B PAM on the expression of mephedrone CPP.  They further investigated the effects of mephedrone on GABA levels in the PFC and hippocampus using two different methods.  They found that baclofen had no effect of expression of mephedrone CPP, although it did reduce activity in the expression test.  The PAM GS39783 did reduce expression of mephedrone CPP.  They showed that mephedrone decreased GABA levels in hippocampus at the dose used for CPP using a chromatographic method, but had no effect in PFC.  Using MRS they found no effect of mephedrone on GABA in the hippocampus.  I have a number of concerns with the manuscript.

  • First, the authors are really studying the effect of the GABA drugs on the memory for a mephedrone CPP and not on mephedrone CPP.  All the central effects of mephedrone they cite in the introduction, such as its effects in the VTA and NAC are not likely to be present during the CPP test when the baclofen drugs are administered.  The hippocampus could be important during the test, but it is unclear to me how that memory effect is specific to mephedrone.  In addition, the one study the authors cite related to GS39783 states that this drug blocked the acquisition, but not expression of nicotine CPP.  It would have made more sense to me to test the GABA drugs on acquisition.

We acknowledge Reviewer’s opinion. Nevertheless, our study primarily focused on evaluating the impact of baclofen and GS39783 on the expression of mephedrone-induced conditioned place preference (CPP). This paradigm is wildely used to study rewarding effects of drugs of abuse (Tzschentke 2007). The brain stuctures maliny involved in this process are ventral tegmental area (VTA) and nucleus accumbens (NAC) and our intention was to provide a broader background on the pharmacological actions of mephedrone in the central nervous system to support the rationale for studying its rewarding effects. We believe that investigating the role of GABA in mephedrone-induced CPP provides valuable insights into the neurochemical mechanisms underlying its rewarding properties.

Also, we would like to state that we agree with the Reviewer’s opinion that studying the effects of drugs on the expression of CPP is highly connected with the memory effect. However, since we also studied the changes in GABA level in the hippocampus (a region involved in memory formation), the choice of the expression-based paradigm was intentional. We acknowledge that the study we cited related to GS39783 and nicotine CPP demonstrated its blocking effect on acquisition. Considering this finding, an acquisition-based experimental design could have provided further insights into the role of GABA-ergic modulation in mephedrone CPP and will be considered in our future projects.

Furthermore, we also would like to highlight that the presented study is a continuation and serves as a complement to our already published data on the relationship between mephedrone effects and glutamatergic system, where an analogous methodology has been successfully used (Wronikowska et al. 2021).

Taking into account the above-mentioned rationale, we believe that the choice of methodological design is well justified.

Wronikowska O, Zykubek M, Michalak A, Pankowska A, Kozioł P, Boguszewska-Czubara A, Kurach Ł, Łazorczyk A, Kochalska K, Talarek S, Słowik T, Pietura R, Kurzepa J, Budzyńska B. Insight into Glutamatergic Involvement in Rewarding Effects of Mephedrone in Rats: In Vivo and Ex Vivo Study. Mol Neurobiol. 2021 Sep;58(9):4413-4424. doi: 10.1007/s12035-021-02404-y. Epub 2021 May 21. PMID: 34021482; PMCID: PMC8487417.

  • Another concern is the reversal from a preference to what looks like a significant aversion after treatment with GS39783 on the CPP test.  I don’t know what that means for in terms of interpretation of the drug effect or even how that could occur.  Any thoughts?

The involvement of the GABA-ergic system in mephedrone-induced reward suggests an interaction between GABA and other neurotransmitter systems implicated in reward and aversion, such as dopamine, serotonin, and noradrenaline. GS39783, as a positive allosteric modulator of GABAB receptors, could potentially influence the balance of these neurotransmitter systems, leading to the observed aversion or preference.

An aversion has been observed for the largest dose of GS39783 (5 mg/kg) (significant difference between mephedrone-conditioned and saline-conditioned animals treated with GS39783 (5 mg/kg) on the test day). However, for the medium dose of GS39783 (2.5 mg/kg), a significant reduction in place preference has been observed without an aversion (no difference between mephedrone-conditioned and saline-conditioned animals treated with GS39783 (2.5 mg/kg)). Therefore, this dose can be perceived as the most prominent and would be used in our future studies.

A rationale for this is present in the Discussion section:

„Interestingly, for a higher dose of GS39783 (5 mg/kg) an aversive behavior has been shown. We may hypothesize that this effect might have been seen due to the inhibitory effect of GABA VTA neurons on DA VTA neurons [54], which may result in a decrease in DA level and place aversion [55]. At the same time, a medium dose of GS39783 (2.5 mg/kg) showed a classic inhibition of mephedrone-induced CPP without any aversive effect and for this reason this dose should be perceived as the most prominent in the presented study.

Some more minor concerns:

  • The scales for the preference test are different for the two drugs.  These scales should be the same.  The use of different scales makes the baclofen effect look small.

The graphs have been reedited according to the Revierwer’s suggestion.

  • Why weren’t the lower doses of mephedrone tested in the CPP tests?  If you want to make comparison to the neurochemical work, using the same doses would have been helpful.  The largest effects on chromatography were at 5 mg/kg, but that dose wasn’t tested on behavior.  Maybe there was an increase in locomotion at that dose.

The lower doses of mephedrone have been tested in the CPP test and already published (Wronikowska et al. 2021a). The rewarding effect has been observed for the doses of 10 and 20 mg/kg. Such effect has not been observed for the dose of 5 and 30 mg/kg. Furthermore, when tested 24h after last mephedrone injection, none of the tested doses (5-30 mg/kg) led to an increase in animals’ locomotion. Detailed explanation of this is also present in the above-mentioned publication. As mephedrone at the dose of 20 mg/kg produced the most robust response in our previous work, this dose has been chosen for the presented study (and also for already published analogous glutamate experiments (Wronikowska et al. 2021b).

Wronikowska O, Zykubek M, Kurach Ł, Michalak A, Boguszewska-Czubara A, Budzyńska B. Vulnerability factors for mephedrone-induced conditioned place preference in rats-the impact of sex differences, social-conditioning and stress. Psychopharmacology (Berl). 2021a Oct;238(10):2947-2961. doi: 10.1007/s00213-021-05910-y. Epub 2021 Jul 15. PMID: 34268586; PMCID: PMC8455394.

Wronikowska O, Zykubek M, Michalak A, Pankowska A, Kozioł P, Boguszewska-Czubara A, Kurach Ł, Łazorczyk A, Kochalska K, Talarek S, Słowik T, Pietura R, Kurzepa J, Budzyńska B. Insight into Glutamatergic Involvement in Rewarding Effects of Mephedrone in Rats: In Vivo and Ex Vivo Study. Mol Neurobiol. 2021b Sep;58(9):4413-4424. doi: 10.1007/s12035-021-02404-y. Epub 2021 May 21. PMID: 34021482; PMCID: PMC8487417.

  • What about anesthetic effects on the imaging data?  Is there data to support that isoflurane had no effect?

The available data generally suggest that isoflurane anesthesia does not significantly alter GABA levels in the brain, including the hippocampus. For example, studies using MRS have shown that isoflurane anesthesia does not significantly affect GABA concentrations in the rat brain, including the hippocampus (Hasselbalch et al., 2008; Zheng et al., 2008). Furthermore, in order to reach reliable comparability, control groups have been subjected to identical conditions and anaesthesia as mephdrone-treated groups.

  • The specificity of the effects to mephedrone needs to be addressed.  Would the GABA drugs have the same effect on other drug or non-drug CPP?

To determine the generalizability of studied GABA drugs' effects on different CPP models, some most prominent data were summarized in the Disucussion section (for both baclofen and GS39783 effects):

„Baclofen has been used as a tool drug in addiction studies for over two decades [30, for review] and has been shown to reduce reinforcing and rewarding effects of cocaine [31-32], nicotine [33-34], heroin [35-36], amphetamine [37] and alcohol [38-39] in several behavioral assays in rodents. Furthermore, it has been reported that pre-treatment with baclofen reduced nicotine-, morphine-, and cocaine-evoked DA release in the shell of the NAc [40]. Interestingly, baclofen has also been used in treatment of alcohol dependence or alcohol use disorders in humans [41]; however, data are insufficient to elaborate on this subject.”

(…)

„GS39783 is a promising agent repeatedly used in research on drug abuse in animal models. GS39783 has been shown to attenuate alcohol-induced increase in locomotion and potentiate the induction of locomotor sensitization [44], as well as to inhibit alcohol self-administration [45-47] and to suppress the acquisition of alcohol drinking behavior [48]. The administration of GS39783 has also attenuated co-caine-induced hyperlocomotion [49] and reduced its reinforcing effects in self-administration [50] and intracranial self-stimulation (ICSS) [32]. Furthermore, it has successfully blocked the expression of amphetamine- [50] and methamphetamine-induced CPP [51]. Lastly, GS39783 was able to decrease nicotine self-administration [52] and to block the acquisition (but not expression) of nicotine-reward in the CPP paradigm [53].”

Based on the above-mentioned evidence it may be concluded that the studied GABA drugs (baclofen and GS39783) have been widely proven to be involved in the reward of different drugs of abuse. Therefore, those agents were chosen to study the involvement of GABA-ergic system in mephedrone effects in our study. Importantly, (answering the Reviewer second question on non-drug CPP), in our study we studied an acute effects of baclofen and GS39783 on the expression of mephedrone-CPP. Therefore, the possibility of the impact of those drugs by themselves on the CPP is excluded due to the applied methodology and acute injections of baclofen and GS39783 on the post-conditioning test. Furthermore, statistical analysis of the baclofen-CPP and GS39783-CPP did not show any differences in saline-conditioned rats.

  • What would it mean for GABA-B receptors specifically if GABA levels are increased in the hippocampus?

Increased GABA levels in the hippocampus may suggest enhanced GABA-ergic signaling in this region. This can potentially lead to activation of GABA-B receptors in the hippocampus. Activation of GABA-B receptors can result in inhibitory effects, such as hyperpolarization of postsynaptic neurons and inhibition of neurotransmitter release, ultimately impacting the overall excitability and function of the hippocampus  such as synaptic plasticity, learning, memory, and mood regulation, reward which are known to involve the hippocampus. Our study was the first one to evaluate whether mephedrone in any way affects GABA levels and/or GABA-dependent effects. As in the presented data there is a clear evidence that GABA-ergic system is invloved in mephedrone effects, in our future project we aim to investigate the specificity of changes in GABA-ergic signaling in the most prominent brain regions.  

Reviewer 2 Report

The introduction provides an interesting and detailed context, clearly articulating the significance of the issue and its relevance. Synthetic cathinones and their effects on the brain, especially the mephedrone drug, are adequately highlighted. This clearly sets the stage for the presented research. The problem at hand is well defined - the unknown relation between the GABA-ergic neurotransmission system and the rewarding effects of mephedrone.

Key concepts are well explained, which would help non-expert readers to understand the paper better. The explanation of the roles of the GABA, GABAA, and GABAB receptors in the brain, and how they are implicated in various disorders, including drug abuse, is particularly well done.

It's impressive to see that the authors' hypothesis is well grounded on the basis of prior studies. Also, the inclusion of the authors' prior research on glutamate involvement in mephedrone-induced reward strengthens the rationale for the current investigation.

The methodology to test the hypothesis, including in vivo and ex vivo determination of GABA hippocampal levels, magnetic resonance spectroscopy, and ion-exchange chromatography, is explained with enough detail in the introduction. Mentioning the use of baclofen and GS39783, substances that interact with GABAB receptors, to investigate the effects on mephedrone-induced conditioned place preference (CPP) also presents the approach of the study clearly.

However, there are some minor suggestions to improve the clarity and readability of this introduction:

A minor note on style - there is some repetition that could be reduced. For instance, the abbreviation for γ-aminobutyric acid (GABA) is repeated multiple times. Once the abbreviation is defined, the full term does not need to be repeated.

In line 54, the phrase "remained under-explored" could be rewritten as "has been under-explored," to better match the tense used throughout the rest of the introduction.

It would be beneficial to briefly explain the term "GABA/glutamate equilibrium" (line 87) for the readers who may not be familiar with it.

The information about the role of GABAB receptors in drug abuse disorder (line 95-96) could be moved up to where GABA receptors are first introduced to give it a bit more context.

While the study mentions the involvement of mephedrone on DA, NA, and 5-HT, it might be beneficial to specify the nature of this involvement – i.e., whether it enhances or inhibits their activity.

Overall, the introduction is well structured, detailed, and provides a solid foundation for the rest of the paper.

The study presents a comprehensive evaluation of the role of the GABA-ergic system in the rewarding effects of mephedrone in rats. The authors leveraged an integrative approach consisting of behavioral, chromatographic, and MRS assessments to identify and assess the impact of GABAB receptors on mephedrone-induced CPP and GABA concentration in the hippocampus.

The rationale for focusing on the GABAB receptors is sound. The authors suggest these receptors as likely molecular targets for substance use disorder treatments, given the limitations associated with GABAA receptor agonists. Two agents, baclofen (a GABAB receptor agonist) and GS39783 (a PAM of GABAB receptors), are employed to evaluate the effects of mephedrone, and the choice is well justified with existing literature on their effects on drug-induced reward.

The findings obtained using GS39783 are promising, confirming the involvement of GABAB receptors in the rewarding effects of mephedrone. The dose-dependent response and the lack of myorelaxant activity associated with GS39783 suggest that PAMs of GABAB receptors could be a safer alternative to orthosteric binding ligands.

However, the results with baclofen are not as clear. Although a decrease in the animals’ locomotor activity due to baclofen could explain the lack of its effect in blocking mephedrone-induced CPP, more investigation is needed to confirm this. Additionally, the reduction of GABA concentration in the hippocampus after mephedrone administration, despite not fully confirming the role of GABAB receptors, still supports their possible involvement.

The study also raises questions about the role of the PFC in mephedrone-induced reward. Future research may need to investigate the effect of mephedrone on the PFC and potential interactions with other neurotransmitters.

The authors acknowledge that their research could have benefitted from studying sex-dependent changes, which would have indeed added more depth to their study. Another limitation is the batch-wise performance of the CPP test, which might have introduced batch effects. 

The authors performed a batch experiment design, adjusting the doses of the tested compounds based on the results of the previous batch. Control animals were included in each batch to maintain the reliability of results. GABA concentrations were measured in the hippocampus and PFC (Prefrontal Cortex), and the authors mention that investigating other areas involved in the drug reward system, like the VTA (Ventral Tegmental Area) or NAc (Nucleus Accumbens), might yield additional informative results.

A noteworthy point was the lack of observed changes in GABA concentration in MRS (Magnetic Resonance Spectroscopy). Despite using a protocol intended to achieve a high signal-to-noise ratio, the authors did not observe the changes they expected. They contemplated using other techniques like MEGA-PRESS or J-resolved methods but decided against it due to potential compromise of results.

One enigmatic observation was the lack of changes in GABA hippocampal level following the administration of mephedrone at the dose of 10 mg/kg. The authors acknowledge this area requires further study to fully comprehend the relationship between mephedrone administration and GABAergic transmission in the hippocampus.

The authors concluded by highlighting the novel findings their research offers. Specifically, they discovered a link between the mephedrone-induced decrease in GABA hippocampal concentration and a behavioral effect where the positive allosteric modulator (PAM) of GABAB receptors, GS39783, blocked mephedrone-induced CPP (Conditioned Place Preference). This suggested that the rewarding effects of mephedrone are partially mediated through GABAB receptors. The authors also argue that these findings could indicate potential therapeutic approaches for substance use disorder, suggesting that the PAMs of GABAB receptors could be useful in drug abuse-related research.

This research also supports previous findings that both GABAergic and glutamatergic systems are implicated in mephedrone-induced reward. This could provide new insights into therapeutic options for mephedrone abuse and addiction. It also suggests potential involvement of these systems in the mechanisms of action for other cathinone derivatives. However, the authors do recognize the need for more research to confirm these suggestions, acknowledging that due to the vast number of new psychoactive substances (NPS) in the cathinone derivatives group, it may not be feasible to test each one, and in many cases, knowledge gained from studying mephedrone could be extrapolated to other cathinone derivatives.

Overall, this study presents important findings in the understanding of the GABA-ergic system's role in mephedrone-induced reward, but future work will be needed to further confirm these findings and explore the mechanisms involved.

Author Response

Reviewer #2

  • The introduction provides an interesting and detailed context, clearly articulating the significance of the issue and its relevance. Synthetic cathinones and their effects on the brain, especially the mephedrone drug, are adequately highlighted. This clearly sets the stage for the presented research. The problem at hand is well defined - the unknown relation between the GABA-ergic neurotransmission system and the rewarding effects of mephedrone.

Key concepts are well explained, which would help non-expert readers to understand the paper better. The explanation of the roles of the GABA, GABAA, and GABAB receptors in the brain, and how they are implicated in various disorders, including drug abuse, is particularly well done.

It's impressive to see that the authors' hypothesis is well grounded on the basis of prior studies. Also, the inclusion of the authors' prior research on glutamate involvement in mephedrone-induced reward strengthens the rationale for the current investigation.

The methodology to test the hypothesis, including in vivo and ex vivo determination of GABA hippocampal levels, magnetic resonance spectroscopy, and ion-exchange chromatography, is explained with enough detail in the introduction. Mentioning the use of baclofen and GS39783, substances that interact with GABAB receptors, to investigate the effects on mephedrone-induced conditioned place preference (CPP) also presents the approach of the study clearly.

We would like to thank the Reviewer for a detailed, comprehensive and in depth revision of our manuscript and for the positive feedback regarding our work.

However, there are some minor suggestions to improve the clarity and readability of this introduction:

  • A minor note on style - there is some repetition that could be reduced. For instance, the abbreviation for γ-aminobutyric acid (GABA) is repeated multiple times. Once the abbreviation is defined, the full term does not need to be repeated.

The repetitions of the abbreviations have been carefully checked. The abbreviation for γ-aminobutyric acid (GABA) has been firstly introduced in current line 54 and has not been repeated afterwards.

  • In line 54, the phrase "remained under-explored" could be rewritten as "has been under-explored," to better match the tense used throughout the rest of the introduction.

Corrected according to the Reviewer’s suggestion.

  • It would be beneficial to briefly explain the term "GABA/glutamate equilibrium" (line 87) for the readers who may not be familiar with it.

According to the Reviewer’s suggestion, a brief clarification has been added to the Introduction:

„Based on the results obtained in our previous studies on glutamate involvement in mephedrone-induced reward [13] and the knowledge of GABA/glutamate equilibrium, we hypothesized that GABA-ergic neurotransmission can be also involved in rewarding effects of this drug. Ample evidence has been provided that the GABA-ergic system is involved in the action of different drugs of abuse and that drugs of abuse can disrupt GABA/glutamate equilibrium by increasing glutamate (excitatory) transmission and decreasing GABA (inhibitory) transmission (see Discussion section).”

Detailed explanation is provided in the Discussion section.

  • The information about the role of GABAB receptors in drug abuse disorder (line 95-96) could be moved up to where GABA receptors are first introduced to give it a bit more context.

According to the Reviewer’s suggestion, the information cited below was moved higher (lines 62-65).

“Notably, available data suggest that GABAB receptors are the ones that are vastly involved in drug abuse disorder [17-19] and, importantly, that GABAA receptors may not be crucial in the development of mephedrone-induced reward [20].”

  • While the study mentions the involvement of mephedrone on DA, NA, and 5-HT, it might be beneficial to specify the nature of this involvement – i.e., whether it enhances or inhibits their activity.

This information has been highlighted in the Introduction:

„Mephedrone exhibits its effects by increasing the concentration (and therefore, enhancing the activity) of dopamine (DA), noradrenaline (NA) and serotonin (5-HT) in the central nervous system (CNS), mainly due to the interaction with plasma membrane transporters for monoamines [4-8].”

  • Overall, the introduction is well structured, detailed, and provides a solid foundation for the rest of the paper.

We would like to thank for this positive feedback.

  • The study presents a comprehensive evaluation of the role of the GABA-ergic system in the rewarding effects of mephedrone in rats. The authors leveraged an integrative approach consisting of behavioral, chromatographic, and MRS assessments to identify and assess the impact of GABAB receptors on mephedrone-induced CPP and GABA concentration in the hippocampus.

The rationale for focusing on the GABAB receptors is sound. The authors suggest these receptors as likely molecular targets for substance use disorder treatments, given the limitations associated with GABAA receptor agonists. Two agents, baclofen (a GABAB receptor agonist) and GS39783 (a PAM of GABAB receptors), are employed to evaluate the effects of mephedrone, and the choice is well justified with existing literature on their effects on drug-induced reward.

The findings obtained using GS39783 are promising, confirming the involvement of GABAB receptors in the rewarding effects of mephedrone. The dose-dependent response and the lack of myorelaxant activity associated with GS39783 suggest that PAMs of GABAB receptors could be a safer alternative to orthosteric binding ligands.

We would like to thank for this in depth summary and positive feedback.

  • However, the results with baclofen are not as clear. Although a decrease in the animals’ locomotor activity due to baclofen could explain the lack of its effect in blocking mephedrone-induced CPP, more investigation is needed to confirm this. Additionally, the reduction of GABA concentration in the hippocampus after mephedrone administration, despite not fully confirming the role of GABAB receptors, still supports their possible involvement.

Aware of the fact that the lack of the observed baclofen effect in the CPP was most likely due to its impact on animals’ mobility, we decided to use another agent (GS39783) to confirm our hypotheses. The rationale behind the lack of the observed baclofen effect and the selection of another compound is present in the Discussion section:

“In our research baclofen showed a dose-dependent effect in mephedrone-conditioned animals; however, statistically it failed to block mephedrone-induced CPP. Nevertheless, the underlying cause of lack of this blockage is most likely attributed to baclofen-induced decrease in animals’ locomotor activity. In order to make sure that the lack of baclofen effect in CPP has been caused by the myorelaxation (which led to the received data distribution), we performed a post-hoc power analysis that showed a non-satisfactory level of power for baclofen post-treatment and interaction of baclofen post-treatment and mephedrone pre-treatment. With an increase of group size, baclofen effect would probably reach statistical significance; however an increase of sample size was not possible due to the ethics reasons in animals’ research. Since baclofen is clinically widely used as a myorelaxant [42]; we were aware that reduction in rats’ mobility might occur, nevertheless based on numerous studies published on baclofen use in drug abuse research, as well as in drug-addicted people, we have decided to administer this compound. Although in this study we have reported mephedrone-induced reduction in GABA con-centration in the hippocampus, based on baclofen-related findings we could not fully confirm the involvement of GABAB receptors in rewarding effects of mephedrone. Furthermore, its therapeutic potential in drug abuse disorders is somewhat limited by its myorelaxant and sedative properties [42-43]. Therefore, to properly assess GABAB receptors' involvement in rewarding effects of mephedrone, we have also tested PAM of these receptors, GS39783”.

  • The study also raises questions about the role of the PFC in mephedrone-induced reward. Future research may need to investigate the effect of mephedrone on the PFC and potential interactions with other neurotransmitters.

We agree with the Reviewer that our study only shed light on the involvement of PFC in mediating mephedrone effects and that further, more detailed, studies are needed in order to fully explain the observed effects. Aware of that, we transparently put this information in the paragraph of Discussion that explained possible liminations of studies:

Furthermore, the evaluation of GABA central concentrations has been performed in the hippocampus and PFC. Presumably, the assessment of GABA level in other areas of drug reward system (such as VTA or NAc) would add an informational value to the presented research”

  • The authors acknowledge that their research could have benefitted from studying sex-dependent changes, which would have indeed added more depth to their study. Another limitation is the batch-wise performance of the CPP test, which might have introduced batch effects. The authors performed a batch experiment design, adjusting the doses of the tested compounds based on the results of the previous batch. Control animals were included in each batch to maintain the reliability of results.

Although the experiments have been performed in batches, as the Reviewer accuretely noticed (and as it is mentioned in the manuscript), we put all the efforts  in order to make the experimental conditions equal for each batch. In the case of our study, the “completely randomized” design would not have been possible due to the large number of animals used. The CPP test is a highly time consuming experiment and it is not possible to increase the number of animals used in the same batch without affecting the quality of the produced data (experiments were conducted between 8.30 a.m. and 4.30 p.m. and prolonging this time would have negatively affected animals’ behavior and therefore the experiment outcome).          

In this case the “randomized block” design has been chosen due to the fact that after performing the first batch the decision has been made on whether to administer higher/lower doses of tested compounds. At the beginning of the experiments we were not entirely sure which doses of baclofen/GS39783 would be found effective. As we did not want to use more animals than necessary (3R rule), we decided to firstly test the most promising dose (based on literature and our preliminary data) and then we decided whether to use higher/lower dose of the tested drugs. As we were aware that testing different doses without adding control animals to each of the batches would not be reliable, we decided to divide control animals and assign them to all of the performed batches.

Furthermore, we would like to state that we made every effort to make the comparison between the batches possible: (1) the experiments were performed by the same investigators and all investigators were assigned to the same responsibilities in all batches); (2) in order to make the conditions as equal as possible, all batches that involved the same tested drug (baclofen or GS39783) were performed one after another (immediately after finishing one batch, the second one has been started); (3) all batches were tested in identical designs (same time of experiments, same weight/age of the animals, same time of handling, same technique of injections performed by the same investigators (and even the same cosmetics used in the shower before entering the experimental zone); (4) all experiments were conducted at the Experimental Medicine Centre in Lublin where animals are reared under specific pathogen-free conditions (SPF) and restrictive rules are implemented (for both maintenance of the animals and conducting experiments) in order to provide the best possible conditions for behavioral testing (http://www.omd.umlub.pl/en/).

Altogether, all measures were undertaken in order to minimize the impact of different variables that could  have influenced observed results.

  • GABA concentrations were measured in the hippocampus and PFC (Prefrontal Cortex), and the authors mention that investigating other areas involved in the drug reward system, like the VTA (Ventral Tegmental Area) or NAc (Nucleus Accumbens), might yield additional informative results.

As the Reviewer accurately noticed, we are aware of this fact and that is why this information has been already presented as a limitation in our manuscript.

  • A noteworthy point was the lack of observed changes in GABA concentration in MRS (Magnetic Resonance Spectroscopy). Despite using a protocol intended to achieve a high signal-to-noise ratio, the authors did not observe the changes they expected. They contemplated using other techniques like MEGA-PRESS or J-resolved methods but decided against it due to potential compromise of results.

The hypotheses behind the lack of the observed effect have been widely discussed in the Manuscript. Furthermore, in order to properly investigate studied effects, the alternative method (chromatographic analysis) has been introduced.

„The lack of observed changes may be due to signal overlap from more abundant metabolites like creatine at 3 ppm, glutamine and glutamate, often denoted as Glx, at 2.3 ppm and N-acetyl aspartate at 2 ppm [70]. As GABA exists in the rodent brains in relatively low concentration (less than 2 mM in the hippocampus) [71] and presents broad peaks with low amplitude in the MRS spectrum. Results obtained in concentration analysis may likely be contaminated by signals from different neighboring compounds [72]; however the effect of signals overlap is less pronounced in higher fields due to inter alia greater spectral resolution, and frequency range [72-74]”.

(…)

„The protocol applied in our study was chosen in order to achieve a high signal-to-noise ratio (SNR) and obtain reliable results from all metabolites existing in the spectrum. On the one hand, it could be worth considering applying GABA-edited solutions like MEGA-PRESS [75] or J-resolved [76] method. On the other hand, we took into account the fact that the adjustments mentioned above might as well compromise the results.”

  • One enigmatic observation was the lack of changes in GABA hippocampal level following the administration of mephedrone at the dose of 10 mg/kg. The authors acknowledge this area requires further study to fully comprehend the relationship between mephedrone administration and GABAergic transmission in the hippocampus.

Indeed, the lack of the observed changes in GABA hippocampal level following the administration of mephedrone at the dose of 10 mg/kg (using both, MRS and chromatographic analysis) is thought-provoking. As mentioned in the Discussion, without further analysis, this effect cannot be fully explained. Nevertheless, in our previous studies regarding glutamate involvement in mephedrone effects, we observed similar phenomenon where in vivo MRS showed that 6 days of mephedrone administration increased glutamate hippocampal level in two time points of measurements: 24 h (for the doses of 5 mg/kg, 10 mg/kg and 20 mg/kg) and 2 weeks (for doses of 5 mg/kg and 20 mg/kg) after last mephedrone injection (Wronikowska et al. 2021). Therefore, the changes in glutamate level after mephedrone (10 mg/kg) were not observed after 2 weeks. The previously published data (Wronikowska et al. 2021) and the data in the presented manuscript clearly suggest an enigmatic effect for this dose of mephedrone that requires further clarification. Experiments that aim to explain this phenomenon have been planned for our future project. Information about the need of further clarification is present in the limitations part of the Discussion:

„Finally, we failed to observe changes in GABA hippocampal level following the administration of mephedrone at the dose of 10 mg/kg in both of the applied analytical methods. Without additional analysis this enigmatic effect remains in the sphere of speculation. Therefore, further studies are of great importance in order to fully explain the rela-tionship between mephedrone administration and GABA-ergic transmission in the hippocampus”.

Wronikowska O, Zykubek M, Michalak A, Pankowska A, Kozioł P, Boguszewska-Czubara A, Kurach Ł, Łazorczyk A, Kochalska K, Talarek S, Słowik T, Pietura R, Kurzepa J, Budzyńska B. Insight into Glutamatergic Involvement in Rewarding Effects of Mephedrone in Rats: In Vivo and Ex Vivo Study. Mol Neurobiol. 2021 Sep;58(9):4413-4424. doi: 10.1007/s12035-021-02404-y. Epub 2021 May 21. PMID: 34021482; PMCID: PMC8487417.

  • The authors concluded by highlighting the novel findings their research offers. Specifically, they discovered a link between the mephedrone-induced decrease in GABA hippocampal concentration and a behavioral effect where the positive allosteric modulator (PAM) of GABAB receptors, GS39783, blocked mephedrone-induced CPP (Conditioned Place Preference). This suggested that the rewarding effects of mephedrone are partially mediated through GABAB receptors. The authors also argue that these findings could indicate potential therapeutic approaches for substance use disorder, suggesting that the PAMs of GABAB receptors could be useful in drug abuse-related research.

This research also supports previous findings that both GABAergic and glutamatergic systems are implicated in mephedrone-induced reward. This could provide new insights into therapeutic options for mephedrone abuse and addiction. It also suggests potential involvement of these systems in the mechanisms of action for other cathinone derivatives. However, the authors do recognize the need for more research to confirm these suggestions, acknowledging that due to the vast number of new psychoactive substances (NPS) in the cathinone derivatives group, it may not be feasible to test each one, and in many cases, knowledge gained from studying mephedrone could be extrapolated to other cathinone derivatives.

As here there are no Reviewer’s comments to address, we just would like to thank for this detailed summary of our conclusions.

  • Overall, this study presents important findings in the understanding of the GABA-ergic system's role in mephedrone-induced reward, but future work will be needed to further confirm these findings and explore the mechanisms involved.

We would like to thank the Reviewer for this comprehensive and detailed revision of our manuscript and for all positive comments on our manuscript. We agree with the Reviewer that the results presented in this manuscript will have to be complemented with the future experiments in this field. For this reason, in the manuscript we transparently explained all identified limitations of our study. Thanks to the received fruitful comments our future experiments will be duly planned in order to fully explain the analysed subject.

Round 2

Reviewer 1 Report

While I appreciate the author’s argument that the study of the expression of mephedrone-induced reward may be an important area of study, the presentation still does not reflect the true nature of the study.  At the very least, the authors need to make clear that expression of reward is what they are studying.  In reading the title, abstract and introduction it would be difficult for the reader to understand the distinction between acquisition and expression.  For example, the title should really be “Relationship between GABA-ergic system and expression of mephedrone-induced reward….”.  While they do mention expression in the abstract, the aim of the study as expressed does not mention expression.  The authors need to be careful in how they talk about reward vs expression of effects based on previous reward.  The effects of the GABA pretreatments in the current study are not directly on mephedrone reward, but the expression of previous mephedrone reward.  The authors should also make clear, that any effect of the pretreatments could not interact directly with the effects of mephedrone.

Author Response

  • While I appreciate the author’s argument that the study of the expression of mephedrone-induced reward may be an important area of study, the presentation still does not reflect the true nature of the study.  At the very least, the authors need to make clear that expression of reward is what they are studying.  

We would like to thank the Reviewer for the insightful comments. We agree with the Reviewer that the „mephedrone-reward” and/or the „expression of mephedrone-reward” should be used carefully in order not to confuse the Readers. We made every effort to clearly show that what we study in the presented study is the expression of mephedrone-reward, rather than mephedrone reward itself. Therefore, according to the Reviewer’s suggestions, we introduced the below-listed changes in the manuscript (highlighted using 'track changes’ tool).

  • In reading the title, abstract and introduction it would be difficult for the reader to understand the distinction between acquisition and expression.  (…) While they do mention expression in the abstract, the aim of the study as expressed does not mention expression.  The authors need to be careful in how they talk about reward vs expression of effects based on previous reward.  

The manuscript has been checked and all misleading information about studying the mephedrone reward has been reedited and changed to studying the expression of mephedrone reward (lines: 2, 17, 24, 97, 110, 141, 177, 229, 264, 295, 391, 415, 417, 425)

In order to make sure the the Readers understand the study design from the very begining, an explanatory sentence has been added to the Introduction:

„The above-mentioned blockage of the expression of previously acquired mephedrone-CPP has been assessed by administering studied GABA-ergic compounds acutely before the post conditioning test.” (lines: 106-108)

Furthermore, the study design is also graphically presented in Figure 1.

  • For example, the title should really be “Relationship between GABA-ergic system and expression of mephedrone-induced reward….”.  

The title has been changed according to the suggestion: ‘Relationship between GABA-ergic system and the expression of mephedrone-induced reward in rats – behavioral, chromatographic and in vivo imaging study’

  • The effects of the GABA pretreatments in the current study are not directly on mephedrone reward, but the expression of previous mephedrone reward. 

  • As mentioned in the answer for the second bullet point, manuscript has been checked and the information about studying the expression of mephedrone reward (rather than mephedrone reward itself) has been highlighted where possible (lines: 2, 17, 24, 97, 110, 141, 177, 229, 264, 295, 391, 415, 417, 425). Furthermore, time of administration of GABA-ergic compounds has been highlighted:
  • In the Introduction: „The above-mentioned blockage of the expression of previously acquired mephedrone-CPP has been assessed by administering studied GABA-ergic compounds acutely before the post conditioning test.” (lines: 106-108);
  • In the Materials and Methods: „In order to evaluate the impact of studied GABA-ergic compounds on the expression of mephedrone-induced reward, on the last day of the CPP procedure, animals underwent a post-conditioning test. Saline (control groups), baclofen (0, 1 or 3 mg/kg) or GS39783 (0, 1, 2.5 or 5 mg/kg) were acutely injected 30 min before the test.” (lines: 517-521);
  • Is presented in Figure 1.

  • The authors should also make clear, that any effect of the pretreatments could not interact directly with the effects of mephedrone.

We understand the rationale behind this comment and, therefore, respecting Reviewer’s suggestion, we decided to add an explanatory paragraph to the Discussion:

„What is worthmentioning is that studied GABA-ergic compounds have been administered acutely before the post-conditioning test, which was performed 24 h after last mephedrone injection. The literature data from animal models suggest that (a) the half-time (t ½) of mephedrone after intravenous (i.v.) injection is short and equals 0.37 h and (b) mephedrone is undetactable in plasma after 9 h after oral administration [56]. Although in our study, mephedrone was administered in i.p. injections, we may hypothesized that the concentrations is plasma would be similar to the above-mentioned i.v. route of administration. Therefore, we may conclude that neither baclofen nor GS39783 (administsred 24 h after last mephedrone injection) interacted directly with mephedrone; however, GS39783 was able to block the expression of previously acquired mephedrone-CPP presumably due to mephedrone-induced changes in central GABA concentration level.” (lines 310-320).

We also would like to once again thank for the insightful revision of our manuscript. We made every effort to improve the quality of the manuscript in order to make it meet the criteria for publication.
